# Enhanced Immunomodulation, Anti-Apoptosis, and Improved Tear Dynamics of (PEG)-BHD1028, a Novel Adiponectin Receptor Agonist Peptide, for Treating Dry Eye Disease

**DOI:** 10.3390/pharmaceutics15010078

**Published:** 2022-12-26

**Authors:** In-Kyung Lee, Kyung-Chul Yoon, Seong-Soo Kang, Su-Kyung Seon, Kwanghyun Lee, Brian B. Kim

**Affiliations:** 1EncuraGen, Inc., Anyang 14058, Republic of Korea; 2Department of Ophthalmology, Research Institute of Medical Sciences, Chonnam National University Medical School and Hospital, Gwangju 61469, Republic of Korea; 3Biomaterial R&BD Center, Chonnam National University, Gwangju 61186, Republic of Korea; 4Department of Ophthalmology, National Health Insurance Service Ilsan Hospital, Goyang-si 10444, Republic of Korea

**Keywords:** dry eye disease (DED), anti-inflammation, epithelial cell protection, anti-apoptosis, (PEG)-BHD1028, adiponectin, peptide drug, AdipoR1, AdipoR2, adiponectin receptors

## Abstract

Dry eye disease (DED) is characterized by impaired tear dynamics, leading to complex pathophysiological conditions. (PEG)-BHD1028, a peptide agonist to AdipoRs, was evaluated as a potential therapeutic agent for DED based on the reported physiological function of adiponectin, including anti-inflammation and epithelial protection. Therapeutic effects of (PEG)-BHD1028 were evaluated in experimentally induced EDE with 0.001%, 0.01%, and 0.1% (PEG)-BHD1028 in mice and 0.1%, 0.2%, and 0.4% in rabbits for 10 days. In the rabbit study, 0.05% cyclosporine was also tested as a comparator. The results from the mouse study revealed significant improvement in tear volumes, tear breakup time (TBUT), inflammation, and corneal severity score (CSS) within 10 days at all (PEG)-BHD1028 concentrations. In the rabbit study, the tear volume and TBUT significantly increased in (PEG)-BHD1028 groups compared with vehicle and 0.05% cyclosporine groups. The CSS, apoptosis rate, and corneal thickness of all (PEG)-BHD1028 and 0.05% cyclosporine groups were significantly improved relative to the vehicle group. The immune cell counts of 0.2% and 0.4% (PEG)-BHD1028 treated groups were significantly lower than those of the vehicle group. These results represent the potential of (PEG)-BHD1028 as an effective therapeutic agent for DED.

## 1. Introduction

Dry eye disease (DED), also known as keratoconjunctivitis sicca, is a multifactorial disease associated with the loss of homeostasis of tear dynamics, which is characterized by tear film instability and hyperosmolarity, inflammation, and damage of the mucosal portion of the eye, and neurosensory abnormalities [1,2]. DED is a growing health issue that affects a patient’s quality of life across physical, psychological, and social dimensions, and DED prevalence ranges from 5% to 50% worldwide [3]. If not adequately treated, DED results in impaired vision or blindness. Several therapeutic agents have been used to treat DED depending on the causes and symptoms, including anti-inflammatory drugs, artificial tears, punctual plugs, and secretagogues. However, treatment adherence is poor because of delayed effectiveness or side effects. Considering the complex multifactorial pathophysiology of DED, an ideal medicine should comprise a more comprehensive therapeutic spectrum.

Adiponectin, a 30-kDa protein adipokine, plays crucial roles in regulating various physiological processes, including improved insulin sensitivity, metabolic homeostasis, anti-apoptosis, and anti-inflammation in selected tissues by binding to its receptors, AdipoR1, AdipoR2, and T-cadherin [4]. Recently, several studies have suggested that adiponectin can provide potential therapeutic opportunities for treating various eye diseases based on their biological functions [5,6,7]. The adiponectin receptors, AdipoR1 and R2, are widely distributed in the lacrimal and conjunctival glands [6]. Sluch et al. reported the presence of AdipoR1 in the neural retina and retinal pigment epithelium and retinal degeneration in *Mfrp^rd6^* and AdipoR1 knockout mice, suggesting a critical role of AdipoR1 in the retinal functions [8]. Topical administration of adiponectin significantly improved clinical signs and inflammatory conditions in a DED mouse model [6]. Bora et al. reported that adiponectin inhibits choroidal angiogenesis or choroidal neovascularization (CNV) in a laser-induced mouse model [9]. (PEG)-BHD1028, a novel 15-aa peptide agonist to AdipoR1 and R2, exhibited its effectiveness in improving insulin resistance, glucose tolerance, anti-apoptosis, anti-inflammatory effect, and energy metabolism through enhanced β-oxidation of fatty acids and mitochondrial biogenesis in vivo and in vitro models [10], thereby suggesting potential therapeutic applications of (PEG)-BHD1028 in DED and other ocular diseases.

(PEG)-BHD1028 was evaluated for its effective dose range and therapeutic efficacy in experimentally induced dry eyes in mice and rabbits. To better understand the sole therapeutic effect of the peptide, avoiding any supplemental effects from excipients or formulation, the pegylated form of the peptide was used by dissolving it in the phosphate-buffered saline. In addition, commercially formulated 0.05% cyclosporine was used to compare the spectrum of the therapeutic effects of (PEG)-BHD1028.

## 2. Results

### 2.1. Baseline Ophthalmic Examinations in Experimentally Induced Dry Eye (EDE) on Day 0

Ophthalmic examinations, including tear volumes, tear film break-up time, and corneal fluorescein staining of healthy (baseline) and EDE animals were performed for grouping before the treatment initiation (Day 0) and revealed no significant discrepancy within each group.

### 2.2. Evaluation of the Tear Production in Each EDE Animal Model

The tear volume in mice was measured using the Phenol Red Thread (PRT) test on days 5 and 10 following the treatment. Tear volume significantly increased in all three (PEG)-BHD1028 treated groups compared to the EDE animals on Days 5 and 10. The tear volume in the 0.01% and 0.1% (PEG)-BHD1028 groups was significantly greater than that of the vehicle group from Day 5 (0.01% group: *p* < 0.01 and 0.1% group: *p* < 0.001 vs. vehicle) and demonstrated a dose-dependent increase (0.01% group: *p* < 0.01 and 0.1% group: *p* < 0.001 vs. 0.001% group) on Day 10 (Appendix A and Figure 1A).

In the study with the rabbit model, a 0.1%, 0.2%, and 0.4% (PEG)-BHD1028 solutions were tested, and 0.05% cyclosporine-treated group was included as a positive reference and comparator. The tear volume, measured using Schirmer’s tear test (STT), significantly increased compared to the vehicle group in all other groups, 0.1% (*p* < 0.001), 0.2% (*p* < 0.05), and 0.4% (PEG)-BHD1028 (*p* < 0.01) groups and 0.05% cyclosporine-treated group (*p* < 0.01) on Day 5 (Appendix A). After 10 days, tear production of all (PEG)-BHD1028 significantly increased compared to the vehicle (*p* < 0.01) (Appendix A and Figure 2A,B). However, the measured tear volume of the 0.05% cyclosporine group on Day 10 did not increase significantly from that on Day 5 or the vehicle group. A comparison between the 0.4% (PEG)-BHD1028 group and 0.05% cyclosporine animals on Day 10 is presented in a separate section along with the details of other testing parameters.

### 2.3. Evaluation of TBUT

Tear film instability is a crucial pathological characteristic of DED and is the most common indicator of tear quality. It is assessed by measuring the time duration until the tear film disruption using fluorescein dye, and the period is recorded as the “tear film break-up time” (TBUT) [11]. In the EDE mice model, TBUT was significantly extended in all (PEG)-BHD1028 groups at *p* < 0.05 compared to the vehicle groups after 10 days of treatment (Appendix A and Figure 1B). The improvement in the 0.01% (PEG)-BHD1028 group was greater than that of the 0.001% (PEG)-BHD1028 group on Day 10 (*p* < 0.05), whereas there was no difference between 0.01 and 0.1% (Figure 1B).

In the EDE rabbit model, the TBUT of all (PEG)-BHD1028 treated groups was significantly greater than that of the vehicle group at *p* < 0.01 and *p* < 0.001, respectively, on Day 5 (Appendix A and Figure 3A). After 10 days, the TBUT of the 0.4% (PEG)-BHD1028 group significantly increased (*p* < 0.01) compared with the vehicle group (Appendix A and Figure 3A). Although not statistically significant, the TBUT of the 0.1%- and 0.2%-treated groups improved relative to the vehicle group or 0.05% cyclosporine group (*p* < 0.058 and *p* < 0.065, respectively) after 10 days of treatments. The TBUT of 0.2% and 0.4% (PEG)-BHD1028 groups exhibited a continually increasing trend over time, whereas the TBUT of the vehicle or 0.05% cyclosporine group tended to stagnate (Figure 3B).

### 2.4. Evaluation of the CFS Score

The clinical severity of the cornea was assessed using corneal fluorescein staining (CFS), a widely used procedure to detect foreign bodies and corneal surface damage. A higher CFS score indicates more severe corneal epitheliopathy. As shown in Appendix A and Figure 4C, the mean CFS score in the 0.1% (PEG)-BHD1028-treated mice group was significantly lower (*p* < 0.05) than in the EDE group after 5 days. After 10 days of treatment, the scores of 0.01% and 0.1% (PEG)-BHD1028-treated animals were significantly lower than that of the EDE (0.01% group: *p* < 0.001 and 0.1% group: *p* < 0.001 vs. EDE) and PBS groups (0.01% group: *p* < 0.001 and 0.1% group: *p* < 0.001 vs. PBS) and were significantly better than that of the 0.001% (PEG)-BHD1028 group (0.01% group: *p* < 0.001 and 0.1% group: *p* < 0.01 vs. 0.001% group) on Day 10.

In the EDE rabbit model, there was no statistically significant difference between the all (PEG)-BHD1028 treated or 0.05% cyclosporine group and the vehicle group after 5 days of treatment. However, after 10 days, the CFS scores significantly decreased dose-dependently in all (PEG)-BHD1028 groups and cyclosporine 0.05% (*p*< 0.01) treated animals relative to the vehicle group (Appendix A and Figure 4A,C). Besides, dynamic slopes of the CFS score change of the 0.2% and 0.4% (PEG)-BHD1028-treated groups from days 1 to 10 were −0.42 and −0.33, respectively, whereas the slope of the 0.05% cyclosporine group was +0.013. These results may imply that a therapeutic effect of 0.2% and 0.4% (PEG)-BHD1028 on improving the clinical severity is better than that of 0.05% cyclosporine (Figure 4B,C).

### 2.5. Anti-Inflammatory Effect

The corneal barrier disruption and epithelial cell loss in DED are accompanied by myeloid and T-cell infiltration [12]. Activated T lymphocytes are the main producers of Th-1 cytokine and interferon (IFN) -γ, which promote cell apoptosis on the ocular surface [13]. To evaluate the anti-inflammatory effect of (PEG)-BHD1028 in the EDE mouse model, immune cells responsible for inflammatory cell surface markers were quantified by flow cytometry, with one animal per group. The CD4 + IFN-γ + -T-cell densities in the cornea of the EDE and vehicle groups were 23.15% and 22.83%, respectively, whereas those in the 0.001%, 0.01%, and 0.1% (PEG)-BHD1028 groups were 18.62%, 16.78%, and 11.66%, respectively. CD4 + IFN-γ + T-cell densities in the conjunctiva of the EDE and vehicle groups were 24.89% and 22.15%, respectively, and those in the 0.001%, 0.01%, and 0.1% (PEG)-BHD1028 groups were 18.96%, 15.31%, and 9.91%, respectively. In the lacrimal gland of the EDE and vehicle groups, the CD4 + IFN-γ + T-cell densities were 21.00% and 22.67%, respectively, and immune cells in the 0.001%, 0.01%, and 0.1% (PEG)-BHD1028 groups were 18.45%, 14.60%, and 10.70%, respectively. (Figure 5A,B). The geometric mean counts of CD11b+ monocytes in the cornea of the EDE and vehicle groups were 298.85 and 237.08, respectively. The counts in the 0.001%, 0.01%, and 0.1% (PEG)-BHD1028 group were 178.06, 131.59, and 126.25, respectively. In the conjunctiva, the CD11b+ monocytes in the EDE and vehicle groups were 550.33 and 547.60, respectively, whereas the counts in the 0.001%, 0.01%, and 0.1% (PEG)-BHD1028-treated groups were 352.09, 330.48, and 36.35, respectively. In addition, in the lacrimal gland, the CD11b+ monocytes in EDE and vehicle groups were 128.63 and 184.62, respectively, and the 0.001%, 0.01%, and 0.1% (PEG)-BHD1028 groups were 84, 74.60, and 60.48, respectively (Figure 5, flow cytometry and Figure 5D, histogram). The results of CD4 + IFN-γ + T cell and CD11b+ cell counts in the cornea, conjunctiva, and lacrimal gland represent the effective anti-inflammatory effect of (PEG)-BHD1028 in the mouse model.

The DED inflammation pathway also includes the infiltration of differentiated (antigen-specific) T cells into the cornea, conjunctiva, and lacrimal glands. To evaluate the anti-inflammatory effect of (PEG)-BHD1028 in the EDE rabbit model, the number of inflammatory cells infiltrated into the stromal layer was measured with H&E staining after 10 days of treatment. The range of the immune cell count in the corneal stroma of six eyes in the normal control group was 273 ± 16.08, 424.78 ± 22.64 in the vehicle group, 399.39 ± 22.22 in the 0.1% (PEG)-BHD1028 group, 349.89 ± 17.18 in the 0.2% (PEG)-BHD1028 group, 305.06 ± 13.88 in the 0.4% (PEG)-BHD1028 group, and 363.07 ± 17.75 in the 0.05% cyclosporine group (Figure 6A,B). The number of immune cells in the corneal stroma of the 0.2% and 0.4% (PEG)-BHD1028 groups were significantly lower than the vehicle group at *p* < 0.05 and *p* < 0.001, respectively. However, there was no significant difference between the 0.05% cyclosporine and the vehicle or 0.1% (PEG)-BHD1028 group. In the conjunctiva, the infiltrated immune cell counts in the normal control group were 415.67 ± 25.01 and 698.78 ± 60.51 in the vehicle-treated group. The counts in 0.1%, 0.2%, and 0.4% (PEG)-BHD1028-treated groups were 624.94 ± 29.29, 472.41 ± 39.06, and 427.22 ± 29.29, respectively. The number of infiltrated immune cells in the cyclosporine 0.05% group was 635.39 ± 50.70 (Figure 6C,D). The immune cells in the conjunctival epithelium of the 0.2% and 0.4% test groups were significantly lower than that in the vehicle-treated group (*p* < 0.001). However, there was no statistically significant difference between the 0.05% cyclosporine and vehicle group. In the lacrimal gland, there was no difference among the study groups (Figure 6E,F).

### 2.6. Corneal Epithelium Thickness Evaluation in the EDE Rabbit Model

The pathophysiology of dry eye disease involves the pathologic changes of the epithelium, such as erosion, scarring, scarring, neovascularization in the cornea and even corneal perforation in severe conditions. The protective effect of (PEG)-BHD1028 on the epithelium was assessed by measuring the corneal thickness using the ZEN (Zeiss, Germany) microscope software after H&E tissue staining. The mean corneal epithelial thickness of six eyes in all (PEG)-BHD1028-treated groups and the 0.05% cyclosporine-treated group was more significant than that in the vehicle-treated animals at *p* < 0.001 (Figure 7A,B). However, there was no significant difference between the (PEG)-BHD1028 and 0.05% cyclosporine-treated groups (Figure 7A,B).

### 2.7. Anti-Apoptotic Effect of (PEG)-BHD1028 in EDE Mouse and Rabbit Models

With DED progression, the epithelium in the anterior portion of the eye is pathologically modified, including functional impairment and structural changes such as secretion of abnormal tear oil and pathogenic apoptosis. Apoptotic cells in the equal area of the cornea and conjunctiva of one eye per group were quantified by the TUNEL assay. As illustrated in Figure 8, (PEG)-BHD1028 exhibited a dose-dependent anti-apoptotic effect on the mouse cornea and conjunctiva.

The percent ratio of apoptotic cells per unit area was evaluated in the rabbit model after 10 days. In the cornea, 35 ± 6% of apoptotic cells were detected in the normal control group and 74 ± 2% in the vehicle group. In the 0.1%, 0.2%, and 0.4% (PEG)-BHD1028-treated groups, the ratio was 40 ± 12%, 39 ± 3%, and 35 ± 4%, respectively. The percent ratio in the cyclosporine 0.05% group was 33 ± 5%. The apoptotic cells in the cornea of all (PEG)-BHD1028 and 0.05% cyclosporine groups were significantly lower than in the vehicle group at *p* < 0.001 (Figure 9A,B). Furthermore, the cell apoptosis percent ratio in the conjunctiva was 7 ± 5% in the normal control group and 56 ± 4% in the vehicle group. In the 0.1%, 0.2%, and 0.4% (PEG)-BHD1028-treated group, the percent ratio was 35 ± 4%, 26 ± 3%, and 19 ± 4% on Day 10. The apoptotic cell percent ratio of the cyclosporine 0.05% group was 32 ± 7% (Figure 9C,D). All tested (PEG)-BHD1028 concentrations and 0.05% cyclosporine significantly reduced corneal and conjunctival apoptosis compared to the vehicle group after 10 days of treatment.

### 2.8. Comparative Analysis between 0.4% (PEG)-BHD1028 and 0.05% Cyclosporine in the Rabbit EDE Model on Day 10

Based on the consistent therapeutic effects of 0.4% (PEG)-BHD1028, all test results of 0.4% (PEG)-BHD1028 and 0.05% cyclosporine on Day 10 were comparatively analyzed using the Student’s *t*-test. The tear volume and TBUT of 0.4% (PEG)-BHD1028 were significantly greater than that of 0.05% cyclosporine (Appendix A). There was no significant difference between CFS scores and immune cell counts of the (PEG)-BHD1028 and the 0.05% cyclosporine groups. However, the mean CFS score of the 0.4% (PEG)-BHD1028 group was approximately 46% less than that of the 0.05% cyclosporine-treated animals (*p* = 0.0558), with a trend of progressive improvement over time. In the immune cell counts, 0.4% (PEG)-BHD1028 exhibited better effectiveness in the conjunctival epithelium (*p* < 0.001) (Appendix A).

## 3. Discussion

The key to treating DED is the restoration of the stability of the ocular surface and control of chronic inflammatory disorders via adequate production of tears with appropriate composition. Although DED is a growing global health issue, no definitive therapy exists. Adiponectin, a 30-kDa cytokine almost exclusively produced by adipocytes, has been known to play significant roles in multiple physio-regulatory processes such as anti-inflammation, insulin sensitization, epithelium protection, and energy metabolism [14,15]. Among many functionalities of adiponectin, numerous studies have shown its anti-inflammatory effects by modulation of immuno-signaling pathways in different cell types [16]. Adiponectin regulates immune responses via the inhibition of proinflammatory factors, such as TNF-α, IL-6, and IFN-γ, promoting macrophage phase change from M1 to M2, and the excretion of anti-inflammatory factors, such as IL-10 and IL-1Rα [14,15,16,17,18,19,20,21,22,23]. Li et al. reported that topical application of globular adiponectin improved clinical signs and decreased inflammation in the ocular surface and lacrimal gland in EDE mice [6]. Besides, several studies have revealed adiponectin’s epithelial cell protective effect [24,25,26,27,28].

(PEG)-BHD1028, a novel peptide agonist to adiponectin receptors, AdipoR1 and R2, was evaluated as a potential therapeutic agent for treating dry eye disease (DED) in experimentally induced dry eye (EDE) animal models based on the reported biological functions of globular adiponectin and the abundance of adiponectin receptors in both anterior and posterior portions of the eye [6,29]. BHD1028 was designed and developed by the authors’ research group based on the novel discovery of the dual binding sites on the receptors [10,30]. (PEG)-BHD1028 is a PEGylated form of BHD1028, a liner 15-mers peptide with the amino acid sequence NH2-Tyr-Tyr-Phe-Ala-Tyr-His-Pro-Asn Ile-Pro-Gly-Leu-Tyr-Tyr-Phe-COOH. Following the peptide synthesis, 5 kDa PEG (polyethylene glycol-5000 Da) was conjugated to the N-terminus to improve the solubility in an aqueous solvent. The PEGylated peptide was evaluated for its ability to activate AMPK against the non-PEGylated form. The in vitro AMPK activation test showed a negligible difference between (PEG)-BHD1028 and non-PEGylated BHD1028 [30]. The biological and physiological characteristics of the peptide were extensively evaluated and verified for its adiponectin-like activities in multiple cellular and disease animal models [10,30].

The primary purposes of these studies were to understand the effective dose level and dosing regimens of (PEG)-BHD1028, as an active ingredient, without other excipients that may extensively affect the effectiveness of the peptide. The peptide solution was prepared in phosphate-buffered saline to fulfill the objectives. However, an appropriate peptide formulation shall be developed to enhance drug delivery to the target tissues and ensure drug stability considering the nature of a peptide molecule and the anatomical features of the eye. The initial concentrations and dose regimens applied to the mice study were determined based on the learnings from various in vitro experiments at various concentrations, in vivo efficacy evaluation in other disease models with the consideration of the potential amount of the drug gets into the eye following the topical instillation. In the rabbit experiment, the dose regimens and concentrations were defined by applying the information obtained from the mice study and anatomical and behavioral differences between the two species, such as the average number of blinks and tissue thickness [31].

In the summary of study results, 0.2% and 0.4% (PEG)-BHD1028 substantially elevated the tear volume and TBUT relative to the vehicle and 0.05% cyclosporine groups. Especially, the 0.4% (PEG)-BHD1028-treated group exhibited significantly increased tear production (*p* < 0.05) and TBUT (*p* < 0.01) compared to cyclosporine on Day 10. Dry eye conditions are largely led by tear film instability, causing physical damage to the ocular surface and secretory glands and the subsequent infiltration of corneal/conjunctival inflammatory cells. The improvement in tear quality and quantity may indicate an improved functionality of gland epithelium tissues, such as conjunctiva, a mucosal tissue rich in goblet cells that maintain the tear stability and homeostasis of the ocular surface [32]. The conjunctival goblet cell density is correlated with clinical severity and is inversely correlated with the cytokine level of ocular surface inflammation [32,33,34]. To better elucidate the relationship between tissue functionality and the quality and quantity of tears following the (PEG)-BHD1028 treatment, tear composition and histo-physiological characterization should be further analyzed.

Along with the improvement in tear dynamics, the CFS scores of animals received 0.2% and 0.4% (PEG)-BHD1028 were significantly lower than that of the vehicle group. The magnitude of the therapeutic effect of 0.4% (PEG)-BHD1028 was markedly more evident than that of 0.05% cyclosporine after 10 days, although the difference was not statistically significant (*p* = 0.0558). In the disease progression analysis, the vehicle group’s CFS score gradually increased over time, and all other groups, except for the 0.4% (PEG)-BHD1028, tended to attenuate the progression after 5 days. In the case of 0.4% (PEG)-BHD1028, however, a continual improvement with a slope of −0.43 was noted.

Tear instability and impaired tear dynamics cause damage to the ocular surface, and these pathophysiological conditions provide an appropriate environment for immune cell infiltration. Animal and cell culture studies demonstrated that IFN-γ reduces conjunctival goblet cell number and, subsequently, mucin production [35]. Both 0.2% and 0.4% (PEG)-BHD1028 showed a significant anti-inflammatory effect in the corneal stroma and conjunctival epithelium compared to the vehicle-treated group. However, there was no difference between the 0.05% cyclosporine and vehicle-treated groups after 10 days. These results correlate well with those of tear production and TBUT. Besides, both (PEG)-BHD1028 and 0.05% cyclosporine showed significant anti-apoptotic effects in the cornea and conjunctiva. The presented data may imply that the reduction of IFN-γ and other immune cells after topical (PEG)-BHD1028 administration eventually suppressed conjunctival goblet cell loss and epithelial apoptosis in an EDE animal model. However, the data also may indicate that the therapeutic scope of cyclosporine is limited to solely anti-inflammatory function.

(PEG)-BHD1028 demonstrated an excellent therapeutic effect on DED with a broader spectrum than 0.05% cyclosporine in EDE animal models. Thus, (PEG)-BHD1028 constituted a novel therapeutic option for DED based on an optimally prepared formulation and dosing regimen, and could thereby overcome the unmet needs of existing medicines.

## 4. Materials and Methods

### 4.1. Synthesis of (PEG)-BHD1028

(PEG)-BHD1028, is a PEGylated form of BHD1028, a linear 15-mer peptide with the amino acid sequence NH_2_-Tyr-Tyr-Phe-Ala-Tyr-His-Pro-Asn Ile-Pro-Gly-Leu-Tyr-Tyr-Phe-COOH. The manufacturing process of (PEG)-BHD1028 consists of two major steps: (1) peptide synthesis and (2) PEGylation and salt exchange. The peptide is synthesized by the Fmoc (9-fluorenylmethoxycarbonyl) solid-phase synthesis process, starting with the swelling and coupling of an anchor resin (H-Phe-2Cl-Trt-Resin). The C-terminus of the first amino acid is coupled with the anchor resin by mixing with HBTU/N- Methylmorpholine/DMF. Then, the Fmoc is deprotected using piperidine/dimethylformamide (DMF). Amino acids are coupled to the growing chain after activating the carboxylic acid terminus. The process is repeated until the last N-terminal tyrosine amino acid is attached. Following the completion of the synthesis, the allyloxycarbonyl (alloc) amino acid protection group on the resin is removed from the peptide by treating it with trifluoracetic acid (TFA). The resulting peptide is precipitated in ether to obtain the crude linear peptide, an intermediate. The intermediate peptide goes through purification and concentration by the reverse-phase HPLC. After the PEGylation with 5 kDa methoxy-polyethylene glycol (mPEG), the crude PEGylated peptide goes through primary purification and recycle purification by RP-HPLC, followed by the acetate exchange using RP-HPLC. The salt-exchange peptide is then lyophilized. To identify the chemical property of (PEG)-BHD1028, the structural confirmation and the intact mass of the API batch were confirmed using FT-IR spectroscopy (FT-IR 4600, JASCO, Japan) (Appendix A) and the MALDI-TOF MS analysis (AB SCIEX TOF/TOF™ 5800, USA) (Appendix A).

### 4.2. Animals

The experimental protocol and animal care complied with the Guide for the Care and Use of Laboratory Animals, and the study was approved by the Institutional Animal Care and Use Committee of the University of Chonnam. The approval date and code of the animal study were CNUHIACUC-21009 on 16 February 2021, for mice and CNU IACUC-YB-2021-123 on 28 September 2021, for rabbits. This study used 8-week-old female C57BL/6 mice and New Zealand white rabbits (Damul Science Inc., Daejeon, Korea) weighing 2 to 3 kg.

### 4.3. Study Design and DED Modeling

Experiments were first conducted in a pilot study in mice to confirm the pharmacological effect and appropriate dose. In addition, the rabbit DED model, in which the ocular surface is largely exposed, was evaluated compared to the mouse model. In the EDE model for mice, DED was induced by subcutaneous injection of scopolamine hydrobromide (Sigma-Aldrich. St. Louis, MO, USA) three times a day (9 AM, 1:30 PM, and 6 PM) with exposure to an air draft in a room at 25 ± 2 °C with an ambient humidity of ˂40%.

The mice were randomly assigned to five groups as follows: (1) EDE (Experimental dry eye) control mice did not receive eye drops, (2) EDE mice treated with vehicle (phosphate-buffered saline), (3) EDE mice treated with 0.001% (PEG)-HD1028, (4) EDE mice treated with 0.01% (PEG)-BHD1028, and (5) EDE mice treated with 0.1% (PEG)-BHD1028. All treatment groups received 2 μL eyedrops three times a day. The rabbit model of DED was induced with twice-daily (8 AM and 4 PM) topical administration of 0.1% BAC (benzalkonium chloride) drops (Sigma, St. Louis, MO, USA). Before BAC treatment, the control groups (baseline, naïve) underwent standard dry eye clinical tests, such as the Schirmer tear test (STT), TBUT, and corneal staining score (CSS). After BAC treatment, dry eye induction was confirmed, and the test substances were administered for 10 days while maintaining dry eye induction. The rabbits were randomly assigned to five groups as follows: (1) EDE rabbits treated with vehicle (phosphate-buffered saline), (2) EDE rabbits treated with 0.1% (PEG)-HD1028, (3) EDE rabbits treated with 0.2% (PEG)-BHD1028, (4) EDE rabbits treated with 0.4% (PEG)-BHD1028, and (5) EDE rabbits treated with 0.05% cyclosporine (Kukjepharm Co., Ltd., Seongnam, Republic of Korea). The testing materials were topically applied BID (bis in die) in both eyes.

### 4.4. Tear Volume Determination

The amount of tear secretion was measured by placing phenol red-impregnated cotton threads (Zone-Quick^TM^; Oasis Medical, Inc., Glendora, CA, USA) in contact with the conjunctival sac on the lateral canthus side of the mouse eye for 20 s, measuring the length using a Nikon SMZ1500 microscope, and then converting the value into the volume by substituting the length into a prescribed formula [36]. Tear production of the rabbit was measured with a modified Schirmer tear test using Whatman 41 filter paper strip (Tianjin Jingming New Technology Development Co., Ltd., Tianjin, China) on days 0, 5, and 10. Eye examination of rabbits was performed under anesthesia. Anesthesia was performed under sedation with an intramuscular injection of xylazine (3 mg/kg, Rompun^®^; Bayer Korea, Republic of Korea), followed by an intramuscular injection of 10 mg/kg tiletamine/zolazepam (Zoletil^®^, Virbac Laboratories, Carros, France). After instilling 1 drop of proparacaine HCl 0.5% (Alcaine^®^, Alcon Korea Ltd., Seoul, Republic of Korea); excess tears and eye drops were removed with Weck-cel^®^ cellulose eye spear. The wetted length (mm) of the paper strip was read after 5 min.

### 4.5. Tear Break-Up Time

TBUT is the time that elapses between the last blink and the first appearance of breakup in the fluorescein and is an indicator of tear film stability. TBUT was assessed by administering 1 μL of 1% sodium fluorescein dye to the lower conjunctival sac of the mouse and visualizing the tear film using cobalt blue light from a slit-lamp biomicroscope. The measurement was repeated three times to obtain an average value.

### 4.6. Evaluating Corneal Fluorescent Staining

Subjects were assessed with corneal staining, and any abnormal findings of the lids, cornea, and conjunctiva were recorded. After instilling 1 μL 1% fluorescent staining solution, washing with saline, and then observing the cornea under a slit-lamp microscope, the degree of epithelial damage (the degree of fluorescence staining) was evaluated by scoring. Corneal staining was graded using the NEI staining grid in which a scale of 0 to 3 (0 = normal, 1 = mild, 2 = moderate, and 3 = severe) was assigned to each of five corneal regions (nasal, central, temporal, inferior, and superior) with a maximum total score of 15 (Ward, 2008).

### 4.7. Evaluation of the Ocular Surface Inflammation in Mice

Flow cytometry was performed to determine the number of the CD4 + IFN-γ + T-cells (anti-CD4 antibody and anti- IFN-γ antibody: BD Biosciences) and CD11b+ cells (anti-CD11b+ antibody: Abcam) from the cornea, conjunctiva, and lacrimal gland of the mouse. Each tissue required for analysis was resected and split with scissors, and then shaken with 0.5 mg/mL collagenase type D at 37 °C for 1 h. After grinding the tissue, the mixture is passed through a cell strainer, centrifuged, and resuspended in phosphate-buffered saline with 1% bovine serum albumin.

### 4.8. Histopathological Evaluation of the Inflammatory Cell Infiltration on the Ocular Surface in Rabbits

The enucleated eyes were washed with phosphate-buffered saline (PBS) and fixed in Modified Davidson’s solution and then embedded in paraffin (Histocore Arcadia, Leica, Wetzlar, Germany). Subsequently, paraffin blocks were sliced into ~6-µm-thick sections, mounted on microscope slides (HistoCore AUTOCUT, Leica, Germany), air-dried and stained with hematoxylin-eosin. The stained tissue slide was imaged, and data were obtained with a light microscope/digital slide scanner (Axio Scan.Z1, Zeiss, Oberkochen, Germany) and image measurements were obtained using ZEN (Zeiss, Germany) and ImageJ (NIH, Bethesda, MA, USA) programs.

### 4.9. Apoptosis Evaluation

In each group, the eye and appendage were excised, fixed in Modified Davidson’s solution, and embedded in paraffin, followed by TUNEL staining of tissue sections. In each section, the degree of apoptosis of the cornea and conjunctiva was measured using ZEN (Zeiss, Oberkochen, Germany) and Image J v1.51j8 software (NIH, Bethesda, MD, USA).

### 4.10. Statistical Analysis

The variability of results was expressed as the mean ± standard error of the mean (SEM) and was considered significant when the *p*-value was <0.05. Statistical analysis was performed using either the Student’s *t*-test or one-way ANOVA followed by Dunnett’s multiple comparisons by using GraphPad Prism 5.0 (GraphPad Software, San Diego, CA, USA).

## Figures and Tables

**Figure 1 pharmaceutics-15-00078-f001:**
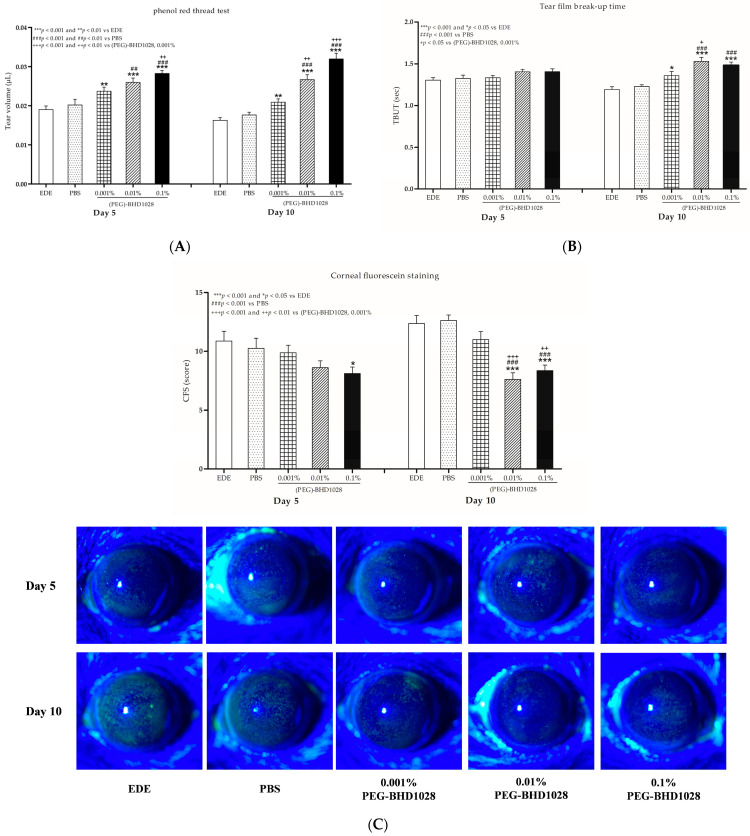
(**A**) The phenol red thread (PRT) test value, (**B**) the tear film break-up time, and (**C**) the corneal fluorescein staining score and representative figures in experimental dry eye (EDE) with PBS, 0.001% (PEG)-BHD1028, 0.01% (PEG)-BHD1028, and 0.1% (PEG)-BHD1028 treated mice. The results of an experiment in both eyes with 4 mice per group are reported. Data are expressed as mean ± SD.

**Figure 2 pharmaceutics-15-00078-f002:**
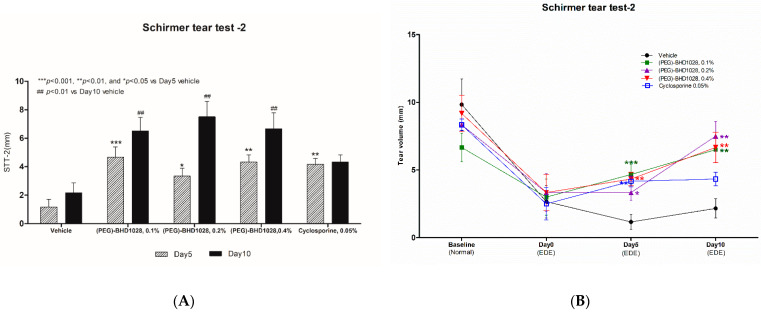
(**A**) The Schirmer tear test (STT) value during induction of dry eye in rabbits. (**B**) The pharmacodynamic trend of tear production. Results are presented as means ± SEM (*n* = 6). *** *p* < 0.001, ** *p* < 0.01, and * *p* < 0.05 vs. Vehicle.

**Figure 3 pharmaceutics-15-00078-f003:**
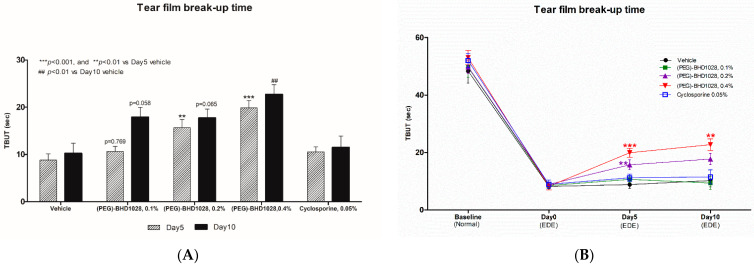
(**A**) The tear film break-up time value during induction of dry eye in rabbits. (**B**) The pharmacodynamic trend of the tear film break-up time. Results are presented as means ± SEM (*n* = 6). *** *p* < 0.001 and ** *p* < 0.01 vs. Vehicle.

**Figure 4 pharmaceutics-15-00078-f004:**
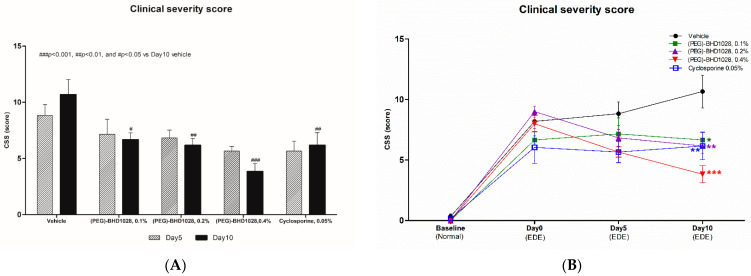
(**A**) The corneal fluorescein staining score during induction of dry eye in rabbits. (**B**) The pharmacodynamic trend of the corneal fluorescein staining score. (**C**) Representative figures of the slit-lamp micrograph of fluorescent staining following corneal damage. Results are presented as means ± SEM (*n* = 6). *** *p* < 0.001, ** *p* < 0.01, and * *p* < 0.05 vs. Vehicle.

**Figure 5 pharmaceutics-15-00078-f005:**
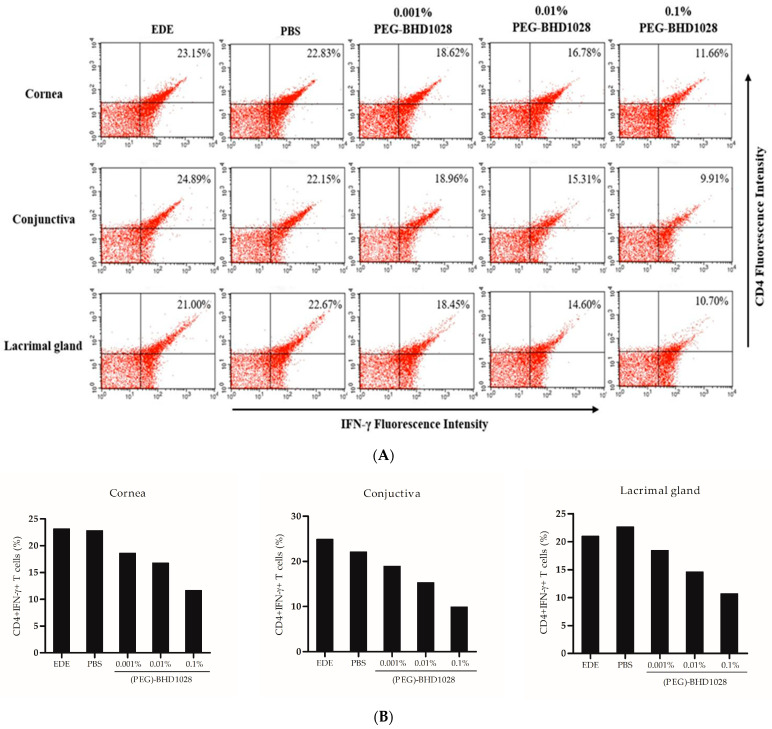
Flow cytometric analysis of CD4 + IFN-γ + T cells (**A**,**B**) and CD11b + T cells (**C**,**D**) in the cornea, conjunctiva, and lacrimal gland of the mice. The results of an experiment used 10,000 cells per group. EDE: experimental dry eye control, PBS: phosphate-buffered saline.

**Figure 6 pharmaceutics-15-00078-f006:**
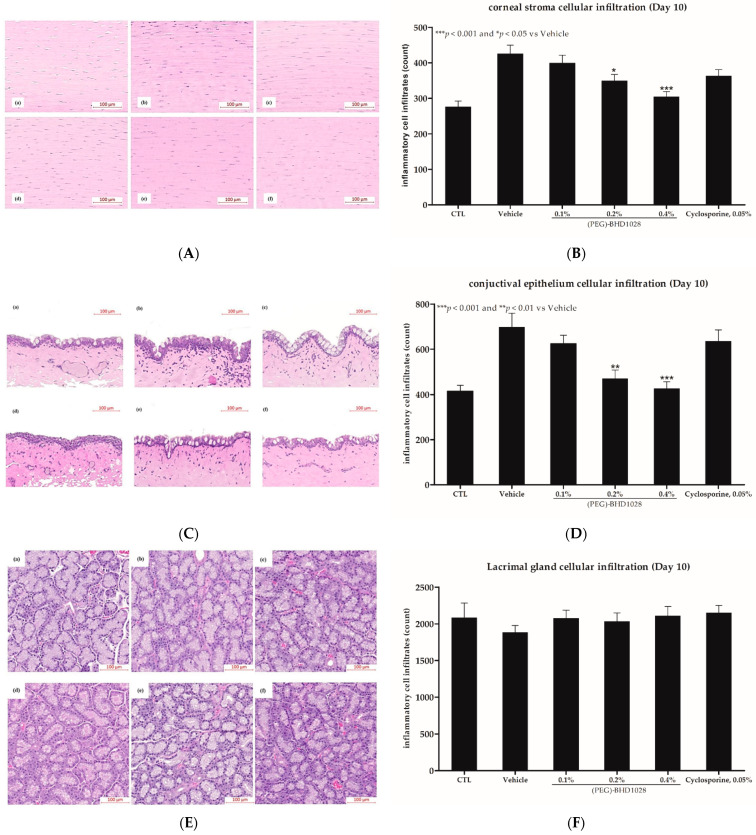
(**A**) Representative images of the corneal stromal immune cell infiltration. (**a**) Control (Normal); (**b**)Vehicle; (**c**) 0.05% Cyclosporine; (**d**) 0.1% (PEG)-BHD1028; (**e**) 0.2% (PEG)-BHD1028; and (**f**) 0.4% (PEG)-BHD1028. (**B**) Data are presented as means ± SEM (*n* = 6). *** *p* < 0.001 and * *p* < 0.05 vs. Vehicle. (**C**) Representative images of the conjunctival epithelial immune cell infiltration. (**a**) Control (Normal) (**b**) Vehicle; (**c**) 0.05% Cyclosporine; (**d**) 0.1% (PEG)-BHD1028; (**e**) 0.2% (PEG)-BHD1028; and (**f**) 0.4% (PEG)-BHD1028. (**D**) Data are presented as means ± SEM (*n* = 6). *** *p* < 0.001 vs. Vehicle. (**E**) Representative images of the lacrimal gland immune cell infiltration. (**a**) Control (Normal); (**b**) Vehicle; (**c**) 0.05% Cyclosporine; (**d**) 0.1% (PEG)-BHD1028; (**e**) 0.2% (PEG)-BHD1028; and (**f**) 0.4% (PEG)-BHD1028. (**F**) Data are presented as means ± SEM (*n* = 6).

**Figure 7 pharmaceutics-15-00078-f007:**
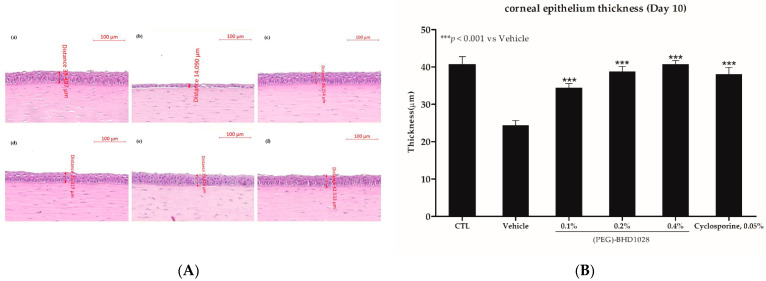
**(A)** Representative corneal epithelial thickness comparison between the normal and substance treatment group in the dry eye model. (**a**) Control (normal); (**b**) Vehicle; (**c**) 0.05% Cyclosporine; (**d**) 0.1% (PEG)-BHD1028; (**e**) 0.2% (PEG)-BHD1028; and (**f**) 0.4% (PEG)-BHD1028. (**B**) Results are presented as means ± SEM. *** *p* < 0.001 vs. Vehicle.

**Figure 8 pharmaceutics-15-00078-f008:**
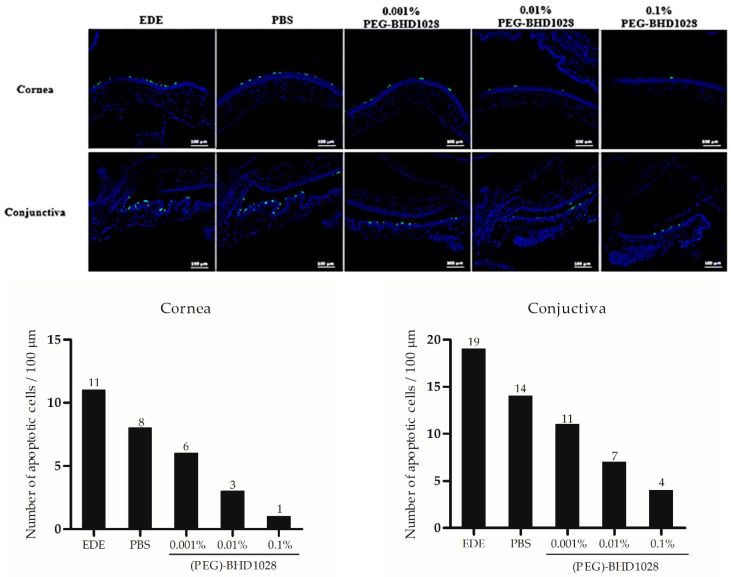
TUNEL staining of representative specimens and apoptotic cell density in mice. Scale bar = 100 μm. EDE: experimental dry eye control, PBS: phosphate-buffered saline.

**Figure 9 pharmaceutics-15-00078-f009:**
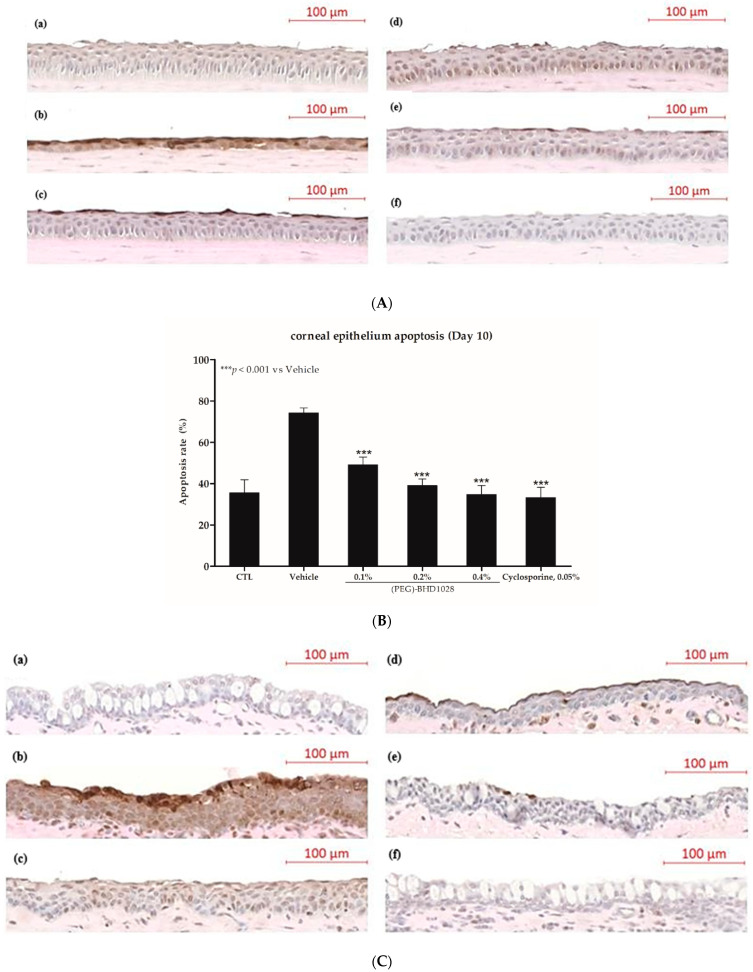
(**A**) TUNEL staining of representative corneal epithelium cells in rabbits. (**a**) Control (normal); (**b**) Vehicle; (**c**) 0.05% Cyclosporine; (**d**) 0.1% (PEG)-BHD1028; (**e**) 0.2% (PEG)-BHD1028; and (**f**) 0.4% (PEG)-BHD1028. (**B**) The apoptotic cell density in corneal epithelium is presented as means ± SEM (*n* = 6). *** *p* < 0.001 vs. Vehicle. (**C**) TUNEL staining of representative conjunctival epithelium cells in rabbits. (**a**) Control (normal); (**b**) Vehicle; (**c**) 0.05% Cyclosporine; (**d**) 0.1% (PEG)-BHD1028; (**e**) 0.2% (PEG)-BHD1028; and (**f**) 0.4% (PEG)-BHD1028. (**D**) The apoptotic cell density in the conjunctival epithelium is presented as means ± SEM (*n* = 6). *** *p* < 0.001 and ** *p* < 0.01 vs. Vehicle.

## Data Availability

The data presented in this study are available in this article (and Appendix A) or can be shared up on request.

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
