# Peer review of "Enhanced Immunomodulation, Anti-Apoptosis, and Improved Tear Dynamics of (PEG)-BHD1028, a Novel Adiponectin Receptor Agonist Peptide, for Treating Dry Eye Disease"

_pharmaceutics, 2022, doi:10.3390/pharmaceutics15010078_

Round 1

Reviewer 1 Report

This study on the potential therapeutic effect of PEG-BHD1028, a peptide agonist to AdipoRs in the treatment of dry eye syndrome is clearly presented in what concerns the methods, their description and analysis of data. The most common used eye drops designated to treat dry eye disease therapeutics include in their composition hyaluronic acid or trehalose that could prevent the water loss from the tear film. The authors considered in this study the potential use of pectin that can bind to the receptors of adinopectin – pectin that is found in lachrymal glands.

In order to prove the efficiencies of the treatment of DEDwith the new peptide compared with the cyclosporine based eye drops, the authors monitored the evolution of classical tests: tear volumes, tear film break-up time, and corneal fluorescein staining of healthy (baseline) and eye dry examination. The results proved a better the anti-inflammatory effect of PEG-BHD1028 compared with cyclosporine. On the other hand, the effects of treatment with cyclosporine and PEG-BHD1028 were similar in what concerns the corneal epithelial thickness. The same observation was noticed in the level of cell apoptosis following the treatment with PEG-BHD1028 and cyclosporine.

The authors also underline that PEG-BHD1028 can be a better candidate for a dry aye drops formulation compared with cyclosporine which has mainly an anti-inflammatory effect. The authors can also look for further studies on mixing this peptide with hyaluronic acid or trehalose.

In general the figures are clearly presented. I suggest for Figure 2 B, Figure 3 B and Figure 4B to change the colours of symbols in such a way to differentiate better (e.g the grey triangle to be replaced by blue and the green square by magenta square).

The authors should indicate in the material section the provenience of the PEG-BHD1028 or to give some information on the synthesis. 

Author Response

Please, check the attached file. 

Reviewer 2 Report

The research presented here on the potential of a PEGylated peptide as a therapeutic for dry eye disease is interesting. However, the manuscripts layout and the varying level of detail reduces the impact of the study and as such I feel the manuscript needs major revision. My comments are as follows:

1) The authors should reconsider the word 'superior' or make a stronger argument for this point. The authors show that the peptide may reduce the rate of apoptosis better than cyclosporine but do not show this statistically. The rest of the comparative data looks like the peptide at 0.4% is comparable to 0.05% cyclosporine. 

2) The authors have published a number of papers on the use of short peptides for adiponectin targeting. It would be convenient for readers to have the peptide sequence shown, structure shown (linear or cyclic), where PEG has been conjugated to the sequence and what (if any) the impact that PEGylation has had on AdipoR binding. 

3) The manuscript moves repeatedly between species (rabbit and mouse) across many figures. This creates some confusion when looking at tear film data etc. and having to remain aware of the species. I highlight this as species blinking rate and tear volume are a critical aspect in this study. I suggest collecting all the mouse data together then following with rabbit data. 

4) The figure captions need significantly more detail. For example, Figure 5. It is unclear from the caption how many cells were used in the flow cytometry, what species the tissue derived from (5C), what cells were selected from the gating (5A) etc. This limited information aspect repeats for a number (but not all) figures. The authors should address this. 

5) The animal eye images should be improved with arrows (or another method) for highlighting key differences for more general readers. Figure 1C particularly could be improved upon (the image looked of low resolution on my pdf). 

Author Response

Please, check the attached file. 

Round 2

Reviewer 2 Report

The authors have addressed my comments and my only minor comment is that the methodology for the synthesis of PEG-BHD1028 should be more detailed or referenced out if published elsewhere. Details should include resin type, solvents, deprotection steps used, PEG conjugation methodology and how the peptide was analysed to confirm structure. Analysis should highlight relevant techniques and briefly data acquired (e.g. if Maldi-ToF MS was used then m/z etc.). 

Author Response

Please, check the attached file.
